# Ethical, Legal, and Social Assessment of AI-Based Technologies for Prevention and Diagnosis of Rare Diseases in Health Technology Assessment Processes

**DOI:** 10.3390/healthcare13070829

**Published:** 2025-04-04

**Authors:** Pietro Refolo, Costanza Raimondi, Violeta Astratinei, Livio Battaglia, Josep M. Borràs, Paula Closa, Alessandra Lo Scalzo, Marco Marchetti, Sonia Muñoz-López, Laura Sampietro-Colom, Dario Sacchini

**Affiliations:** 1Department of Health Care Surveillance and Bioethics, Section of Bioethics and Medical Humanities, Università Cattolica del Sacro Cuore, 00168 Rome, Italy; costanza.raimondi1@unicatt.it; 2Research Centre for Clinical Bioethics & Medical Humanities, Università Cattolica del Sacro Cuore, 00168 Rome, Italy; 3Melanoma Patient Network Europe (MPNE), Fjällbo Selknä 152, 75597 Uppsala, Sweden; violeta.astratinei@mpneurope.org; 4Asociatia Melanom Romania (AMER), 050663 Bucharest, Romania; 5National Agency for Regional Health Services (AGENAS), 00187 Rome, Italy; battaglia@agenas.it (L.B.); loscalzo@agenas.it (A.L.S.); marchetti@agenas.it (M.M.); 6Department of Clinical Sciences, University of Barcelona, 08036 Barcelona, Spain; jmborras@iconcologia.net; 7Fundació de Recerca Clínic Barcelona, Institut d’Investigacions Biomèdiques August Pi i Sunyer, 08036 Barcelona, Spain; closa@clinic.cat (P.C.); munoz6@recerca.clinic.cat (S.M.-L.); lsampiet@clinic.cat (L.S.-C.)

**Keywords:** health technology assessment (HTA), artificial intelligence (AI), rare diseases, prevention, diagnosis, ethics, legal, social

## Abstract

Background: While the HTA community appears well-equipped to assess preventive and diagnostic technologies, certain limitations persist in evaluating technologies designed for rare diseases, including those based on Artificial Intelligence (AI). In Europe, the EUnetHTA Core Model^®^ serves as a reference for assessing preventive and diagnostic technologies. This study aims to identify key ethical, legal, and social issues related to AI-based technologies for the prevention and diagnosis of rare diseases, proposing enhancements to the Core Model. Methods: An exploratory sequential mixed methods approach was used, integrating a PICO-guided literature review and a focus group. The review analyzed six peer-reviewed articles and compared the findings with a prior study on childhood melanoma published in this journal (Healthcare), retaining only newly identified issues. A focus group composed of experts in ethical, legal, and social domains provided qualitative insights. Results: Thirteen additional issues and their corresponding questions were identified. Ethical concerns related to rare diseases included insufficient disease history knowledge, lack of robust clinical data, absence of validated efficacy tools, overdiagnosis/underdiagnosis risks, and unknown ICER thresholds. Defensive medicine was identified as a legal issue. For AI-based technologies, concerns included discriminatory outcomes, explicability, and environmental impact (ethical); accountability and reimbursement (legal); and patient involvement and job losses (social). Conclusions: Integrating these findings into the Core Model enables a comprehensive HTA of AI-based rare disease technologies. Beyond the Core Model, these issues may inform broader assessment frameworks, ensuring rigorous and ethically responsible evaluations.

## 1. Introduction

In recent years, significant advancements have been made in the development of Health Technology Assessment (HTA) methodologies for the ethical, legal, and social assessment of various types of technologies, including medical and surgical interventions, diagnostics, screening, and complex health technologies [1,2,3]. These developments have strengthened the ability of HTA frameworks to assess the broader impact of health innovations beyond clinical and economic dimensions.

While the HTA community appears well-equipped to assess preventive and diagnostic technologies [4], certain limitations persist in the evaluation of technologies designed for rare diseases [5], including those based on Artificial Intelligence (AI) [6].

In Europe, a well-known example of a framework for the assessment of preventive and diagnostic technologies is the EUnetHTA Core Model^®^ [7] (hereafter referred to as Core Model). Developed by the European Network for Health Technology Assessment (EUnetHTA), this framework supports the production of HTA reports in a standardized and systematic manner. Its primary objective is to provide a structured approach to the evaluation of health technologies in order to facilitate collaboration among HTA agencies across Europe. Key features of this framework include the following:▪ Modular structure: the model is divided into nine domains—health problem and current use, technical characteristics, safety, clinical effectiveness, economic evaluation, ethical analysis, organizational aspects, legal aspects, and patient/social aspects—allowing for flexible and targeted assessments;▪ Standardization: it promotes consistency and comparability in HTA reports across different countries, facilitating knowledge sharing and cross-border applicability;▪ Comprehensive approach: beyond clinical and economic factors, the model integrates ethical, social, and organizational considerations to provide a multidimensional assessment;▪ Adaptability: while structured, it remains flexible to accommodate the specific needs of different healthcare systems and regional contexts;▪ Collaborative tool: it enables joint assessments by multiple HTA agencies, reducing duplication of efforts and fostering international cooperation;▪ Transparency: by clearly defining domains, questions, and methodologies, the model enhances clarity in decision-making and ensures openness in the assessment process.

In essence, the Core Model establishes a structured approach for each domain—such as safety, effectiveness, economic impact, and ethical analysis—by organizing topics into key issues and their corresponding questions. These issues and questions identify key elements that must be considered in the assessment process. By systematically addressing them, evaluators can conduct a comprehensive and methodologically rigorous evaluation of the technology under investigation.

The ethical analysis domain encompasses six topics—benefit–harm balance, autonomy, respect for persons, justice and equity, legislation, and ethical consequences—covering a total of 20 issues. The legal analysis domain consists of seven topics—patient autonomy, patient privacy, equality in healthcare, ethical aspects, authorization and safety, ownership and liability, and market regulation—spanning 18 issues [7]. Finally, the patients and social analysis domain includes three topics—patients’ perspectives, social group aspects, and communication aspects—addressing eight issues.

Although precise quantitative data on the use of the Core Model across Europe remain limited, several key indicators highlight its widespread adoption and integration into HTA processes:▪ Joint Clinical Assessments (JCA): the Core Model serves as the methodological foundation for JCA under the new EU HTA Regulation (EU) 2021/2282, ensuring a systematic evaluation of health technologies across Europe [8];▪ Integration into national HTA processes: several European countries, including Germany, France, Italy, and Sweden, have incorporated the Core Model’s principles and structure into their national HTA frameworks [9];▪ Training and capacity building: regular workshops and training sessions organized by EUnetHTA attracted representatives from various European HTA agencies, demonstrating sustained interest and application [10];▪ Citations in literature and guidelines: the Core Model is frequently referenced in HTA-related publications, best practice documents, and methodological guidelines, underscoring its recognized value [11];▪ Global interest and adaptation: while primarily used in Europe, the model has garnered international attention, with some HTA bodies outside the region exploring its applicability for their own assessments [12].

Despite its considerable adaptability, the Core Model is not well-suited for evaluating technologies designed for rare diseases or those based on AI—two categories that present significant challenges for the HTA community.

Rare diseases, by definition, affect a small number of individuals, posing challenges such as limited clinical data, small patient cohorts, and difficulties in conducting large-scale randomized trials [13]. These constraints complicate the HTA evaluation process, which traditionally relies on robust evidence generation and comparative effectiveness analysis.

Similarly, AI-based technologies, such as those increasingly adopted in dermatology [14], pose unique challenges to HTA methodologies, as they often evolve rapidly, rely on complex algorithms, and require continuous updates.

The aim of this article is to identify key issues and corresponding questions relevant to the ethical, legal, and social assessment of AI-based technologies for the prevention and diagnosis of rare diseases within HTA processes. The ultimate goal is to enhance the Core Model by incorporating a set of specific topics and corresponding issues tailored to the assessment of technologies designed for rare diseases and based on AI.

The article is part of the European Union-funded initiative “Novel health care strategies for melanoma children, adolescents and young adults” (MELCAYA) (https://www.melcaya.eu/, accessed on 10 February 2025) (grant agreement ID 101096667). This initiative aims to expand knowledge on melanoma risk factors and determinants to improve prevention, early detection, and prognosis in CAYA (hereafter referred to as childhood melanoma or CHM). CHM, classified as an ultra-rare malignancy, affects individuals aged 20 years or younger, with an incidence of 1.3–1.6 cases per million in children under 15 and approximately 15 cases per million in adolescents aged 15–19. Notably, its occurrence has been rising at an annual rate of 4.1% among adolescents since 1997 [15]. Despite its rarity, CHM is the most prevalent malignant skin cancer within this age group across the EU-27 countries [16]. Within this context, our unit is focused on developing resources and recommendations to assist health authorities in making informed decisions regarding the early diagnosis and prevention of CHM.

In alignment with the specific objectives of the MELCAYA project, the research question of the present article has been refined into two sub-questions: A. What specific issues, beyond those addressed in the EUnetHTA Core Model^®^, should be considered in the ethical, legal, and social assessment of technologies designed for rare diseases? B. What specific issues, beyond those addressed in the EUnetHTA Core Model^®^, should be considered in the ethical, legal, and social assessment of AI-based technologies?

## 2. Materials and Methods

An exploratory sequential mixed methods design was used for this study [17,18], starting with a literature analysis and followed by a qualitative focus group discussion to identify gaps and validate research findings. This approach facilitated the integration of evidence from existing studies and the firsthand perspectives of experts in the field. By combining insights from existing literature with qualitative findings, this approach provided a comprehensive understanding of both conceptual and practical aspects. The study adhered to the procedural guidelines established by Skamagki et al. [19]. The stepwise process followed in this study is illustrated in Figure 1, which provides a visual representation of the methodology.

### 2.1. Literature Review and Data Synthesis 

A structured PICO (Population, Intervention, Comparator, Outcome) framework guided the literature review to ensure accuracy and alignment with the research objectives. The search was conducted in December 2023 in PubMed without time restrictions, focusing exclusively on peer-reviewed publications in English. Inclusion and exclusion criteria were predetermined and are detailed in the Appendix A.

Two researchers (P.R. and C.R.) independently screened the retrieved studies using Rayyan 2024 (https://www.rayyan.ai/, accessed on 27 December 2024). They assessed the titles and abstracts of publications that met the inclusion criteria. Any ambiguous cases were discussed with a third researcher (D.S.) to reach a consensus. Additionally, manually identified relevant studies were incorporated into the selection process.

To ensure a comprehensive understanding, each selected study was read at least twice before data extraction. Data extraction was conducted independently by two reviewers (P.R. and C.R.) using a standardized template, which captured key details such as the study’s source, focus, overarching considerations, and ethical, legal, and social issues. A consensus meeting was then held to compare extracted data, resolve discrepancies, and refine accuracy through a joint review of the full text. Additional details on the study selection and data extraction process can be found in the Appendix A.

The synthesis of the selected studies was conducted using a critical interpretive synthesis approach [20]. P.R. and C.R. systematically categorized the extracted data into thematic issues aligned with the research questions. A comparative analysis was then conducted between these identified issues and those addressed in the Core Model, retaining only those not already covered.

In the final stage, the identified issues were further compared with those previously outlined in a study published in this journal (Healthcare) [21], which examined the ethical, legal, and social implications of melanoma prevention and diagnosis in CHM. Only issues not identified by the aforementioned study or the Core Model were retained.

Finally, for each issue in the final list, a corresponding question was developed to facilitate the ethical, legal, and social assessment of the technologies, thereby integrating the EUnetHTA framework. The process of deriving questions from the identified issues followed an interpretative approach, ensuring that each question effectively captured the essence of the corresponding issue. Specifically, the goal was to formulate questions that explicitly addressed the critical aspects expressed by each issue, transforming broad concerns into targeted inquiries.

### 2.2. Focus Group Discussion

Following the literature review, a focus group discussion was conducted to gather qualitative insights and validate findings. The session adhered to the Standards for Reporting Qualitative Research (SRQR) guidelines [22] and took place online on 29 January 2025. Six experts from Canada (1), Sweden (1), Italy (1), Norway (1), and the Netherlands (2) participated, all specializing in ethical, legal, and social evaluation within HTA processes.

The recruitment process followed a two-step approach. First, eligible participants were identified from the membership list of the HTAi Interest Group for Ethical Issues in HTA (https://htai.org/interest-group/ethics-in-hta/, accessed on 27 December 2024), facilitated by the involvement of one of the article’s authors (P.R.), who serves as the co-chair of this group. This research group includes individuals engaged in HTA, such as representatives from HTA bodies, industry, government, academia, as well as patients and citizens, all of whom have an interest in the explicit and formal identification and analysis of ethical, legal, and social issues in HTA.

The chosen participants were then sent an invitation letter detailing the purpose of the focus group. This letter included information about the study’s potential benefits and risks, highlighted the voluntary nature of participation, and assured recipients of their right to withdraw at any stage—before, during or after the session. Before the discussion commenced, each participant provided written consent via email.

The 60 min session, conducted in English, was facilitated by two moderators (P.R. and D.S.) and supported by a rapporteur (C.R.). At the outset, the compiled list of issues and questions was presented. The discussion followed the same research questions (A and B) to gather feedback and identify potential gaps. C.R. summarized the key insights from the discussion while maintaining participant anonymity, ensuring that no direct quotations appeared in the final report. After the session, the collected data were systematically analyzed, focusing on the key issues that were effectively addressed. The findings from this analysis are presented in the Appendix A.

## 3. Results

The literature search process employed two complementary strategies, yielding a total of 45 unique citations. After an initial screening based on titles and abstracts, 40 citations were excluded for not meeting the predefined eligibility criteria, leaving five articles for inclusion. Additionally, one relevant article was identified through reference screening, bringing the final total to six included studies.

Among these, two studies [23,24] specifically examined the ethical, legal, and social assessment of technologies dedicated to rare diseases, while the remaining four [25,26,27,28] focused on the ethical, legal, and social assessment of AI-based technologies.

The literature review led to the identification of eight issues and the development of corresponding questions. The comparison with the article published by Refolo [21] on the ethical, legal, and social implications of melanoma prevention and diagnosis in CHM resulted in the identification of four additional issues and questions. Furthermore, the focus group discussion revealed one more issue. In total, 13 issues were identified and their respective questions were formulated.

The following paragraphs present the final outcomes of this integrative process. Where applicable, references to the pertinent literature are provided to ensure that each finding is thoroughly contextualized and substantiated by existing scholarly work. By systematically linking the results to established research, this approach enhances the study’s transparency and reproducibility, ensuring clear traceability of the sources underpinning the conclusions.

### 3.1. A. What Specific Issues, Beyond Those Addressed in the EUnetHTA Core Model^®^, Should Be Considered in the Ethical, Legal, and Social Evaluation of Technologies Designed for Rare Diseases?

Five additional ethical issues and one legal issue were identified, while no new social issues were found. Below is a narrative elaboration of these issues, along with the corresponding questions developed.

#### 3.1.1. Ethical Issues

Insufficient knowledge of the natural history of rare diseases: due to their low prevalence, rare diseases are often poorly characterized in terms of progression, long-term complications, and potential responses to interventions. This limited understanding not only impairs the ability to accurately predict patient outcomes but also hinders the development of targeted interventions for improved disease management [23]. The ethical concern arises from the risk of inappropriate management resulting from the lack of a well-characterized disease trajectory.

Question: Is the natural history of the disease, its progression, and its long-term effects known?

Lack of sufficient and robust clinical data: the inherently small patient populations and the scarcity of large-scale clinical trials for rare diseases result in a limited availability of high-quality, reliable data. This paucity of evidence undermines the ability to conduct rigorous evaluations of new technologies, increasing uncertainty regarding their safety, efficacy, and overall impact on patient outcomes. The absence of standardized pathways makes it particularly challenging to identify appropriate comparators in HTA evaluations, thereby complicating the assessment of alternatives and their relative effectiveness [23,24]. The ethical dilemma lies in balancing the need for rigorous evidence with the urgency of providing timely access to potentially beneficial interventions. The lack of large-scale trials means that regulatory bodies and HTA agencies must make decisions based on limited and sometimes inconclusive data, increasing the risk of approving ineffective or harmful interventions. Conversely, excessively rigid evidence standards may lead to unjustified delays in access to innovative technologies. Ethical tensions between scientific rigor, patient autonomy, and the principle of beneficence are underscored.

Question: Is there any other type of obstacle to evidence generation regarding the benefits and harms of the intervention?

Lack of validated instruments to assess efficacy and effectiveness endpoints: for many rare diseases, universally accepted tools or metrics to measure clinically meaningful outcomes are lacking. The absence of standardized assessment instruments poses a significant challenge in evaluating the efficacy and real-world effectiveness of new technologies, further complicating regulatory and reimbursement decisions [23,24]. The absence of standardized assessment tools raises ethical concerns about the fairness and reliability of HTA evaluations, as decisions on reimbursement and adoption of new technologies may rely on arbitrary or inconsistent criteria. As a result, promising interventions may be excluded simply because they do not align with traditional assessment frameworks, ultimately disadvantaging patients with rare diseases.

Question: Are there well-established instruments or metrics to assess the efficacy and effectiveness of the technology?

Overdiagnosis and underdiagnosis risks: overdiagnosis and underdiagnosis pose significant challenges in the assessment of technologies for rare diseases [29,30,31]. Overdiagnosis may lead to unnecessary treatments, increased healthcare costs, and psychological distress for patients, while underdiagnosis may result in delayed interventions, worsening prognoses, and reduced access to appropriate therapies. In the context of HTA, evaluating technologies for rare diseases requires a careful balance between sensitivity and specificity to mitigate these risks and ensure the effective allocation of healthcare resources.

Question: Does the implementation of technology involve risks of overdiagnosis or underdiagnosis?

Application of incremental cost-effectiveness ratio (ICER) thresholds: traditional cost-effectiveness frameworks, particularly those based on the incremental cost-effectiveness ratio (ICER), may not be well-suited for rare diseases. Given the typically high costs of orphan devices and the small patient populations they serve, standard economic thresholds may systematically undervalue new interventions [23]. This raises ethical concerns regarding equitable access to preventive and diagnostic measures, as well as the broader applicability of conventional HTA methodologies in addressing the unique needs of patients with rare conditions.

Question: Are there any obstacles to evidence generation regarding the economic evaluation of the intervention?

#### 3.1.2. Legal Issue

Defensive medicine risks: defensive medicine poses a significant risk in the context of technologies for rare diseases. To mitigate legal liability, healthcare professionals may overuse or underuse diagnostic tests, prescribe unnecessary procedures, or adopt overly cautious medical practices. While this approach seeks to reduce the risk of misdiagnosis and potential malpractice claims, it can also lead to increased healthcare costs, heightened patient anxiety, and inefficient resource allocation. In the context of HTA, it is crucial to assess how medical technologies influence defensive medical behaviors and to develop strategies that promote evidence-based decision-making while ensuring that necessary interventions are carried out and unnecessary ones are minimized.

Question: Is the implementation of the technology associated with issues related to defensive medicine?

### 3.2. B. What Specific Issues, Beyond Those Addressed in the EUnetHTA Core Model^®^, Should Be Considered in the Ethical, Legal, and Social Evaluation of AI-Based Technologies?

Our research identified three additional ethical issues, two additional legal issues, and two additional social issues. Below is a narrative elaboration of these issues, along with the corresponding questions developed.

#### 3.2.1. Ethical Issues

Discriminatory outcomes: since algorithms inherently reflect the data they are trained on, biased input datasets can directly affect their performance and generalizability. The use of culturally biased health data poses a significant risk, potentially leading to inaccurate or discriminatory outcomes. In dermatology, for example, a biased image dataset may result in misclassification of benign nevi as malignant (false positives), leading to unnecessary biopsies, increased healthcare costs, patient anxiety, and potential physical harm. Conversely, bias in the dataset might also lead to false negatives, where malignant lesions are incorrectly classified as benign, delaying crucial diagnoses and treatment. The ethical concern here revolves around equity and non-discrimination: if AI algorithms are trained on datasets that fail to represent diverse populations, they may reinforce existing biases, exacerbating disparities in healthcare outcomes [24,27]. This issue is particularly critical in HTA processes, where recommendations may inadvertently favor certain demographic groups over others. Ethically, it is imperative to ensure that AI systems undergo rigorous fairness assessments and that mechanisms are in place to identify and mitigate biases before deployment.

Question: Does the implementation or use of the technology lead to discrimination due to biased health data?

Explicability: explicability in AI refers to the ability of AI models to provide clear, understandable, and transparent explanations of how they arrive at their decisions or conclusions. This feature is crucial for enabling users, stakeholders, and regulators to comprehend the reasoning and processes behind AI-based outcomes. If these AI models operate as “black boxes”, with opaque decision-making processes, healthcare professionals and policymakers may struggle to verify, trust, or challenge their recommendations [25,26,27,28].

Question: To what extent can the technology provide interpretable and understandable explanations of the reasoning behind its results?

Environmental impact: AI-based healthcare solutions, particularly those involving machine learning algorithms and large-scale data processing, require substantial computational power. This results in high energy consumption, contributing to carbon emissions and raising concerns about the sustainability of AI applications in healthcare. Given the growing integration of AI in medical decision-making, diagnostics, and administrative processes, its environmental footprint warrants closer scrutiny within HTA frameworks [32,33]. The ethical implications extend beyond immediate healthcare benefits to broader ethical responsibilities, highlighting the need to incorporate sustainability considerations into HTA evaluations of AI-based innovations.

Question: How should the environmental impact of the technology be assessed to ensure that its deployment does not disproportionately burden specific populations, regions, or healthcare systems, particularly in terms of energy consumption, resource allocation, and long-term sustainability?

#### 3.2.2. Legal Issues

Accountability: determining responsibility when an AI system makes a mistake is a complex issue influenced by several factors. Accountability may rest with the developers, the organization deploying the AI, or regulatory bodies overseeing its use. Developers are responsible for designing reliable, safe algorithms and ensuring they are thoroughly tested and free from biases. Organization utilizing AI must ensure it is applied appropriately, within its intended scope, and with proper oversight. In healthcare, when an AI-related mistake causes harm, legal and ethical responsibility may extend across multiple parties, including those who implemented, maintained, or oversaw the AI system. The key ethical issue lies in assigning moral and professional responsibility when AI systems fail or produce harmful recommendations [25,27].

Question: Is accountability clearly defined in the event that the technology makes a mistake?

Reimbursement: the adoption of AI-based tools raises critical questions about reimbursement policies [34]. As these technologies become increasingly integrated into healthcare, it is essential to establish fair and sustainable reimbursement models that promote accessibility and support the implementation of innovative diagnostic methods.

Question: Are the reimbursement policies related to technology implementation well-defined?

#### 3.2.3. Social Issues

Patient information: patients should be informed about the role of AI instruments in their care for several important reasons [27,35]. First, transparency is essential for building trust between patients and healthcare providers. When patients understand that AI is being used to assist in decision-making, they can better comprehend how diagnoses or treatment recommendations are made. Additionally, informing patients about the role of AI allows them to ask questions and seek clarification, particularly if AI plays a significant role in their medical care. Transparency also ensures that patients are aware of AI’s potential limitations, including the possibility of errors or biases, which can impact their healthcare outcomes. Ultimately, educating patients about AI fosters a more collaborative healthcare environment.

Question: When and to what extent will patients be informed about the involvement of AI?

Job loss: the impact of AI on employment remains a topic of ongoing debate among experts [36]. Some believe that AI and automation may displace a substantial number of jobs, particularly those involving routine and repetitive tasks, potentially resulting in significant unemployment in specific industries. Conversely, others believe AI will generate new opportunities, driving innovation and efficiency across industries. In the context of HTA, evaluating the workforce impact of AI technologies is essential for informed policy decisions. This includes assessing how AI reshapes labor distribution in healthcare, identifying potential gaps in workforce competencies, and implementing reskilling and upskilling initiatives.

Question: Is the implementation of the technology linked to risks of job losses?

Table 1 summarizes the results of our research.

## 4. Discussion

Our analysis has identified 13 key issues and corresponding questions that, when integrated with those already present in the Core Model, can enhance the ethical, legal, and social assessment of AI-based technologies for the prevention and diagnosis of rare diseases.

One fundamental aspect we wish to emphasize is that the categorization of various issues as ethical, legal, or social, as well as the formulation of the questions, reflects the interpretation of our research team. For instance, the challenges related to “defensive medicine”, which we have classified as legal in nature, also have ethical implications, particularly concerning patient autonomy. Moreover, some issues that we have identified as novel might be considered by others as already encompassed within the Core Model. For example, the question regarding “explicability” could be seen as falling under the broader issues related to balancing the benefits and risks associated with the implementation of the technology under investigation.

This debate—whether technologies for rare diseases or AI-based technologies truly raise new issues—extends beyond the HTA community. Some question whether the growing number of frameworks for evaluating technologies such as AI-based tools reflects a merely redundancy of analytical approaches [37,38] or a genuine need [39,40]. This concern was also raised during our focus group discussion, where one expert suggested that certain issues might already be incorporated into the Core Model. Despite these divergences, our position aligns with the view that making ethical and methodological concerns more explicit is preferable to leaving them implicit or assumed. Transparency in evaluation processes fosters accountability, enhances stakeholder engagement, and mitigates the risk of overlooking critical ethical dimensions. While we acknowledge the potential for redundancy, we argue that the benefits of increased clarity, rigor, and inclusivity in technology assessment outweigh the drawbacks. In this sense, even if some concerns are already embedded within established frameworks, explicitly articulating them can strengthen the evaluative process and contribute to more informed decision-making [41].

Additionally, the formulation of the questions itself is a matter of interpretation, and alternative phrasings were certainly possible. Our goal was to remain as consistent as possible with the style of EUnetHTA. This alignment was facilitated by the involvement of some of the authors of this article (PR, DS) in the development of the Core Model.

To the best of our knowledge, our framework is the first to simultaneously address both the assessment of technologies for rare diseases and the evaluation of AI-based technologies by integrating insights from both the existing literature and expert opinions. This dual approach allows for a more robust and multidimensional analysis, capturing not only the theoretical and empirical findings documented in prior research but also the practical, real-world perspectives of professionals directly engaged in the evaluation and implementation of these technologies.

Furthermore, beyond the Core Model, the issues and questions we propose can also be used independently; they may serve as a valuable foundation for developing assessment frameworks for technologies with similar characteristics. The flexibility of our approach allows it to be adapted to other emerging health technologies that share ethical, legal, and social complexities, such as personalized medicine, digital health interventions, and novel biotechnological applications. By explicitly articulating key concerns and structuring them within a comprehensive framework, we aim to contribute to the refinement of evaluation processes, ensuring that both the benefits and potential risks of innovative technologies are thoroughly examined. Future research could further build upon our findings by testing the applicability of this framework in real-world decision-making processes, engaging a broader range of stakeholders, and refining its components based on empirical validation.

However, we do not claim that the discussion on the ethical, legal, and social assessment of technologies for rare diseases or based on AI is exhaustive. At present, numerous initiatives at both the European and global levels are addressing these topics. For instance, Health Technology Assessment International (HTAi) (https://htai.org/, accessed on 10 February 2025), the largest international scientific association dedicated to advancing HTA worldwide, hosts two specialized study groups focused on rare diseases and AI. These groups actively contribute to research and discussion in their respective areas. Additionally, the European Union has funded a large-scale initiative called EDiHTA (European Digital Health Technology Assessment) (https://edihta-project.eu/, accessed on 10 February 2025), which aims to develop the first flexible, inclusive, validated, and ready-to-use European framework for assessing digital health technologies (DHTs). This framework will facilitate the evaluation of various DHTs—such as telemedicine, mobile applications (mApps), and AI—at different levels of technological maturity and across different territorial scopes (national, regional, and local), incorporating perspectives from payers, society, and hospitals. Another project funded at the European level is also ASSESS DHT (https://assess-dht.eu/the-project/, accessed on 10 February 2025).

Thus, our work represents just a small contribution to a much broader and ongoing process that will involve the HTA community in the coming years, aiming to establish an initial groundwork for further development. The continuous evolution of healthcare systems, coupled with the increasing complexity of health technologies, calls for dynamic and adaptable HTA frameworks. Moreover, the challenges in this field are set to intensify, particularly as HTA methodologies must align with an evolving legislative landscape.

To cite just two examples in Europe, the recent implementation of Regulation (EU) 2021/2282 on HTA—which establishes a framework for joint clinical assessments and cooperation among Member States—and Regulation (EU) 2024/1689 on AI—which introduces harmonized rules for AI systems, including those applied in healthcare—will significantly shape future evaluations. The fact that our work employs the EUnetHTA Core Model as its foundational framework may be a significant advantage, as this framework is inherently aligned with Europe’s broader effort to standardize and harmonize health technology assessment across Member States.

While our study presents several strengths, we also recognize certain limitations that should be acknowledged.

First, there is a possibility that some relevant articles were inadvertently omitted from our analysis. This is due to the inherent complexity of ethical, legal, and social issues, which often overlap and can be interpreted differently depending on the research framework and disciplinary perspective. As a result, studies addressing closely related yet differently contextualized topics may have been excluded. Moreover, we identified only six relevant articles, a number we acknowledge as relatively low. This limited pool constrains the breadth of our analysis and underscores the scarcity of existing literature on this topic. While this highlights the need for further research, it also emphasizes the preliminary nature of our findings, which should be interpreted with caution. However, the insights gained from our review, supplemented by perspectives from focus group discussions with experts, have likely mitigated the risk of overlooking relevant aspects. Future research could further address this limitation by refining definitional boundaries and employing more precise inclusion criteria, thereby ensuring a more comprehensive and representative analysis of the literature.

Second, the scope of our findings is somewhat constrained by the limited number of experts who participated in the focus groups. This limitation was primarily determined by the specific objectives and methodological framework of the MELCAYA project. Although the qualitative insights obtained are undoubtedly meaningful, the small sample size may affect the applicability of the conclusions. Furthermore, given the rapidly evolving nature of AI-based technologies and rare disease interventions, a larger and more heterogeneous group of experts could provide additional insights into emerging ethical, legal, and social concerns that may not have been fully addressed within our study. The inclusion of professionals from different geographical regions, institutional backgrounds, and areas of expertise—such as policymakers, regulatory authorities, patient advocacy groups, and industry stakeholders—would offer a more holistic perspective and enhance the external validity of the findings. Increasing both the diversity and number of participants in future studies would enhance the reliability of the findings and improve their applicability across various contexts.

Finally, our research is primarily shaped by a European perspective. The Core Model, which served as our main analytical framework, was developed within a European context. However, it represents only one of many internationally established approaches. Notable alternatives include the Technology Appraisal Process of the National Institute for Health and Care Excellence (NICE) in England [42], the methods and guidelines of the Canada’s Drug Agency [43], and the World Health Organization (WHO)’s Health Technology Assessment of Medical Devices [44]. Moreover, all but one of the focus group participants were European. This geographic and cultural focus constitutes an inherent limitation, as it may not fully capture perspectives from other regions. Future research should incorporate a more global outlook, integrating viewpoints from diverse healthcare systems and regulatory environments to provide a more comprehensive assessment of these complex issues.

## 5. Conclusions

Our investigation identified 13 additional key issues that extend beyond the EUnetHTA Core Model for the ethical, legal, and social assessment of AI-based technologies in the prevention and diagnosis of rare diseases. Specifically, five ethical concerns emerged regarding rare disease technologies, namely insufficient knowledge of disease history, lack of robust clinical data, absence of validated efficacy assessment tools, risks of overdiagnosis and underdiagnosis, and the application of ICER thresholds, along with defensive medicine, which was identified as a legal issue.

For AI-based technologies, additional concerns included discriminatory outcomes, explicability, and environmental impact from an ethical perspective; patient involvement and job losses as social challenges; and accountability and reimbursement as legal issues.

These findings highlight the complexity of assessing AI-based health technologies, particularly in the rare disease domain, where conventional evaluation frameworks may not fully capture emerging ethical, social, and legal challenges. A more comprehensive and adaptable approach in future HTA frameworks will be crucial to ensuring that these technologies are assessed in a manner that is both methodologically rigorous and ethically and socially responsible.

## Figures and Tables

**Figure 1 healthcare-13-00829-f001:**
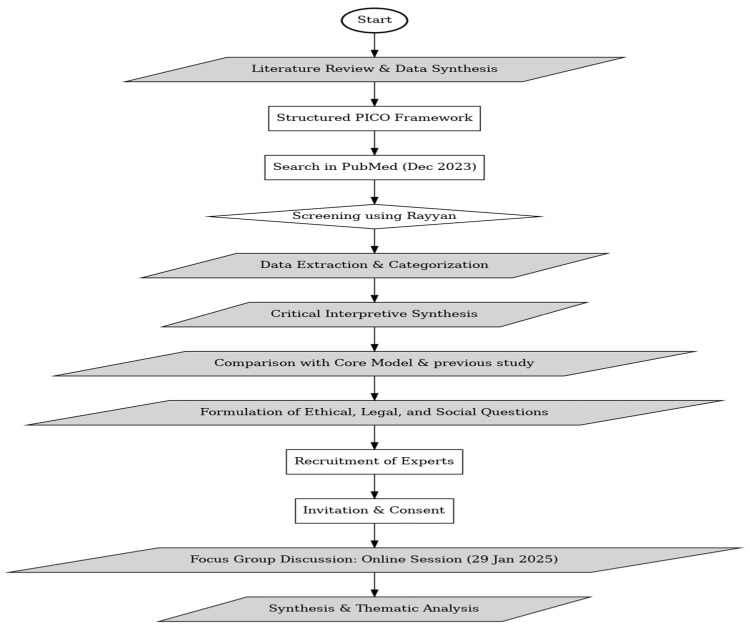
Flowchart illustrating the methodological steps followed in the study.

**Table 1 healthcare-13-00829-t001:** Summary of results.

Type of Issue	Context	Question
Ethical	Rare diseases	Is the natural history of the disease, its progression, and its long-term effects known?
Ethical	Rare diseases	Is there any other type of obstacle to evidence generation regarding the benefits and harms of the in-tervention?
Ethical	Rare diseases	Are there well-established instruments or metrics to assess the efficacy and effectiveness of the technology?
Ethical	Rare diseases	Does the implementation of technology involve risks of overdiagnosis or underdiagnosis?
Ethical	Rare diseases	Are there any obstacles to evidence generation regarding the economic evaluation of the intervention?
Legal	Rare diseases	Is the implementation of the technology associated with issues related to defensive medicine?
Ethical	AI-based	Does the implementation or use of the technology lead to discrimination due to biased health data?
Ethical	AI-based	To what extent can the technology provide interpretable and understandable explanations of the reasoning behind its results?
Ethical	AI-based	How should the environmental impact of the technology be assessed to ensure that its deployment does not disproportionately burden specific populations, regions, or healthcare systems, particularly in terms of energy consumption, resource allocation, and long-term sustainability?
Legal	AI-based	Is accountability clearly defined in the event that the technology makes a mistake?
Legal	AI-based	Are the reimbursement policies related to technology implementation well-defined?
Social	AI-based	When and to what extent will patients be informed about the involvement of AI?
Social	AI-based	Is the implementation of the technology linked to risks of job losses?

## Data Availability

Data supporting the reported results can be found in Appendix A.

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
