# Peer review of "Ethical, Legal, and Social Assessment of AI-Based Technologies for Prevention and Diagnosis of Rare Diseases in Health Technology Assessment Processes"

_healthcare, 2025, doi:10.3390/healthcare13070829_

Round 1

Reviewer 1 Report

Comments and Suggestions for Authors

While the HTA community appears well-equipped to assess preventive and diagnostic technologies, certain limitations persist in evaluating technologies designed for rare diseases, including those based on Artificial Intelligence (AI). In Europe, the EUnetHTA Core Model® serves as a reference for assessing preventive and diagnostic technologies.

AUTHORS aim to identify key ethical, legal, and social issues related to AI-based technologies for the prevention and diagnosis of rare diseases, proposing enhancements to the Core Model.

THEY proposed an exploratory sequential mixed methods approach, integrating a PICO-guided literature review and a focus group.

THEIR review analyzed six peer-reviewed articles and compared the findings with a prior study on childhood melanoma published in this journal, retaining only newly identified issues. A focus group composed of experts in ethical, legal, and social domains provided qualitative insights. Thirteen additional issues and their corresponding questions were identified. Ethical concerns related to rare diseases included insufficient disease history knowledge, lack of robust clinical data, absence of validated efficacy tools, overdiagnosis/underdiagnosis risks, and unknown ICER thresholds. Defensive medicine was identified as a legal issue. For AI-based technologies, concerns included discriminatory outcomes, explicability, and environmental impact (ethical); accountability and reimbursement (legal); and patient involvement and job losses (social). AUTHORS concluded that: (1) integrating these findings into the Core Model enables a comprehensive HTA of AI-based rare disease technologies. (2) Beyond the Core Model, these issues may inform broader assessment frameworks, ensuring rigorous and ethically responsible evaluations.

This is a  very interesting study

I have some minor comments for the authors:

1) The introduction gives a lot of space to the European HTA. I suggest integrating it with a greater look at the realities of other nations.

2) I really like the methods. A structuring of the description of the activities in subsections with a short summary and a flow chart would make them perfect.

3) Results: “The following paragraphs present the final outcomes of this integrative process.

Where applicable, references to the relevant literature are provided to support and contextualize the results.” Give some more information about the structure and the type of analysis you performed

4) The presentation of section 3.1 with questions and answers is interesting, but probably the inclusion of some editorial tool such as a table would help to improve the presentation.

5) Section 3.2: same problem as section 3.1

6) Discussion: expand in line 418 the contribution to the discussion of the individual studies (“approaches [37−40].”)

7) In addition to the three studies cited in the group in line 418 there are no other literature references used to corroborate the study. It is suggested to integrate others and to expand

8) A list of acronyms would be helpful for the reader.

Author Response

We would like to express our sincere gratitude for your thoughtful and detailed review of our manuscript. Your insightful comments and suggestions have been invaluable in improving the quality of our research and enhancing the overall clarity of the paper. We truly appreciate your feedback and hope that the revised version meets your expectations.

We have carefully addressed all the points you raised, and we believe that the revisions have strengthened the manuscript. Please find our detailed responses to each of your comments below.

1) The introduction gives a lot of space to the European HTA. I suggest integrating it with a greater look at the realities of other nations.

Thank you so much for the valuable suggestion. As requested, we have included a reference to other countries in the Discussion.

2) I really like the methods. A structuring of the description of the activities in subsections with a short summary and a flow chart would make them perfect.

Thank you very much for the suggestion. We have divided the methods into subsections and included an illustrative flowchart.

3) Results: “The following paragraphs present the final outcomes of this integrative process.

Where applicable, references to the relevant literature are provided to support and contextualize the results.” Give some more information about the structure and the type of analysis you performed

We apologize if we were not sufficiently clear. We have revised the text, clarifying the criterion we followed.

4) The presentation of section 3.1 with questions and answers is interesting, but probably the inclusion of some editorial tool such as a table would help to improve the presentation.

Thank you very much for the suggestion. We have included a table that summarizes the results of our research (3.1 and 3.2)

5) Section 3.2: same problem as section 3.1

See above

6) Discussion: expand in line 418 the contribution to the discussion of the individual studies (“approaches [37−40].”)

Thank you! We have separated the references to clarify to whom the positions belong.

7) In addition to the three studies cited in the group in line 418 there are no other literature references used to corroborate the study. It is suggested to integrate others and to expand

Thank you! We have integrated the text.

8) A list of acronyms would be helpful for the reader.

The abbreviations are at the end of the article

Reviewer 2 Report

Comments and Suggestions for Authors

The Authors conducted research related to assessing Ethical, legal, and social concepts in AI-based technologies for the prevention and diagnosis of rare diseases in HTA processes. The topic addresses an important subject in a relevant field and proposes outcomes beneficial for the community. However, some issues have to be discussed before further evaluation.

1. The authors indicated that they considered only six studies. How sufficient is that? Please explain.

2- How does this study differ from the similar ones existing in the literature?

3- The question "Is the implementation of the technology linked to risks of job losses?" is not answered or the answer is not provided.

4- Please revise the statement "in this journal" that refers to a study as the phrase "this journal" is not clear.

5- Please include a future directions paragraph indicating what else can be done in the future using the outcomes of this work and considering the limitations of this work.

Author Response

We would like to express our sincere gratitude for your thoughtful and detailed review of our manuscript. Your insightful comments and suggestions have been invaluable in improving the quality of our research and enhancing the overall clarity of the paper. We truly appreciate your feedback and hope that the revised version meets your expectations.

We have carefully addressed all the points you raised, and we believe that the revisions have strengthened the manuscript. Please find our detailed responses to each of your comments below.

  1. The authors indicated that they considered only six studies. How sufficient is that? Please explain.

Thank you very much. We have clarified this weakness more thoroughly in the discussion.

2- How does this study differ from the similar ones existing in the literature?

Thank you very much for the suggestion. We have added text on this point in the discussion.

3- The question "Is the implementation of the technology linked to risks of job losses?" is not answered or the answer is not provided.

We apologize if the text was unclear, but the question regarding the risks of job loss is explicitly stated earlier. The question is the outcome of the expressed issue. We hope that the table included will help provide better clarity.

4- Please revise the statement "in this journal" that refers to a study as the phrase "this journal" is not clear.

Thank you for the suggestion. We have clarified in the text which journal it refers to.

5- Please include a future directions paragraph indicating what else can be done in the future using the outcomes of this work and considering the limitations of this work.

Thank you very much for the suggestion. We have added text on this point in the discussion.

Reviewer 3 Report

Comments and Suggestions for Authors

The manuscript addresses a highly relevant topic in the field of Health Technology Assessment (HTA), focusing on the ethical, legal, and social impact of Artificial Intelligence-based technologies for the diagnosis and prevention of rare diseases. The study stands out for its timeliness, methodological rigor, and innovative contribution to expanding the EUnetHTA Core Model®.

The subject matter is particularly relevant, especially considering the increasing integration of AI in healthcare and the specific challenges in evaluating rare diseases.

The use of an exploratory sequential mixed methods approach is appropriate and well-justified. The integration of a literature review and a focus group strengthens the validity of the findings.

The article follows a well-organized structure, with a clear articulation between the introduction, methodology, results, and discussion.

The identification of 13 new issues and their corresponding guiding questions enriches the assessment framework of the Core Model.

The cited sources are relevant and include recent studies, reinforcing the credibility of the research.

However, there are some areas where the manuscript could be improved:

  • The discussion on the limitations of the focus group (e.g., the small number of participants) is appreciated, but it could be expanded to explore how these limitations may have influenced the results.

Author Response

We would like to express our sincere gratitude for your thoughtful and detailed review of our manuscript. Your insightful comments and suggestions have been invaluable in improving the quality of our research and enhancing the overall clarity of the paper. We truly appreciate your feedback and hope that the revised version meets your expectations.

We have carefully addressed all the points you raised, and we believe that the revisions have strengthened the manuscript. Please find our detailed responses to each of your comments below.

1) The discussion on the limitations of the focus group (e.g., the small number of participants) is appreciated, but it could be expanded to explore how these limitations may have influenced the results.

Thank you very much for the suggestion. We have added text on this point in the discussion.

Round 2

Reviewer 2 Report

Comments and Suggestions for Authors

Thanks to the Authors for considering my comments. The paper can be considered for publication in its current version.

Author Response

Thank you very much. Your suggestions have been very valuable.